

# Cyclones enhance the transport of sea salt aerosols to the high atmosphere in the Southern Ocean

Jun Shi[1,2], Jinpei Yan*[1,2], Shanshan Wang[1,2], Shuhui Zhao[3], Miming Zhang[1,2], Suqing Xu[1,2], Qi Lin[1,2], Hang Yang[1,2], Siying Dai[1,2]

[1]Key Laboratory of Global Change and Marine Atmospheric Chemistry, Ministry of Natural Resources, Xiamen 361005, China;

[2]Third Institute of Oceanography, Ministry of Natural Resources, Xiamen 361005, China.

[3]School of Tourism, Taishan University, Tai'an City, Shandong Province, China, 271000.

**Corresponding author**:

Jinpei Yan, Phone: 86-592-2099290, E-mail: jpyan@tio.org.cn

Address:No. 178Daxue Road, Siming District, Xiamen City, Fujian Province, China

**Abstract**:Cyclones are expected to increase the vertical transport of sea salt aerosols (SSAs), which may significantly impact the climate by increasing the cloud condensation nuclei (CCN)/cloud droplets ($N_d$) population, hence changing the radiation reflected back to space. In this

study, high temporal resolution (1h) aerosol composition measurements were performed during a survey in the southern hemisphere middle and high latitudes during the period 23 February 2018 to 4 March 2018. The characteristics of sea salt aerosols (SSA) during three cyclones were observed during the survey. The level of SSA increased very little with increasing wind speed during the cyclones, which is quite different from the expected case that wind speed will increase SSA

concentration. However, the size of sea salt aerosol particles during the cyclones were larger than during the no-cyclone periods. It seems that the generation of sea salt aerosols is enhanced during cyclones, but the SSA concentration near the sea surface does not increase. Calculations suggest that more than 23% of SSAs can be transported upward during a cyclone period which can result in considerable quantities of SSAs being transported to high altitudes. The upward transport also

lowers the level of SSAs in the lower atmosphere. Additionally, the transport of SSAs to the high atmosphere during cyclones increases the CCN burden in the marine boundary layer. This study extends the knowledge of SSA generation and transport during cyclones, which has implications to climate change.

**Keywords**: Sea salt aerosol, Cyclone, Southern Ocean, transport



## 1. Introduction

Sea salt aerosol (SSA), one of the largest sources of primary aerosols in the marine atmosphere, makes a significant contribution to atmospheric aerosols and the primary inorganic sea salt component of SSAs dominates the marine aerosol mass size distribution (McInnes et al., 1996). It is reported that the annual global SSA flux is estimated to be $1.01 \times 10^4 \text{ Tg yr}^{-1}$ (Gong et al., 2002), and mostly consists of NaCl and a mixture of one or more other salts, such as Mg, K, Ca sulfates, and traces of organic material (Thomas et al., 2022). SSAs are considered to be the most important contributor to aerosol light scattering in the marine boundary layer (MBL) (Quinn and Coffman, 1999; Takemura et al., 2002). In addition, SSAs act as cloud condensation nuclei (CCN) (Pierce and Adams, 2006), thereby altering the reflectivity, lifetime and extent of clouds and hence further influence the global climate.

The influence of SSAs on cloud properties is thought to be particularly strong over remote ocean regions devoid of continental particles. For all supersaturations in the Southern Ocean (SO), SSA makes up more than 20% of the total CCN and up to 65% for low supersaturation, determined using a lognormal-mode-fitting procedure (Quinn et al., 2017). SSAs are formed predominantly by the action of the wind on the ocean (Stokes et al., 2013) as the major mechanism of SSA production is air bubbles bursting at the surface of the ocean as a result of wind stress (Monahan and Muircheartaigh, 1980). Wind stress on the ocean surface forms waves, bubbles are then formed and return to the surface, creating whitecaps and bursts, injecting sea water films and jet droplets into the atmosphere. Wind speed can also significantly affect the size distribution of SSAs (McDonald et al., 1982), although some studies have suggested that wind speed may not be the sole condition affecting SSA production as humidity, temperature and sea-air temperature difference may also





affect their formation (Cole et al., 2003; Shi et al., 2022; Liu et al., 2020). However, the production

of SSAs in extreme weather (such as cyclones) in the southern hemisphere middle and high latitudes,

especially in the SO, still remains uncertain.

The westerlies of the SO are located at middle and high latitudes in the southern hemisphere.

Westerlies fundamentally control regional patterns of air temperature while also regulating ocean

circulation, heat transport, and carbon uptake (Goyal et al., 2021). Moreover, the zone of westerlies

is prone to cyclones which dominate the precipitation pattern of the southern hemisphere mid-

latitudes (Mycoy et al., 2020). The SO plays an important role in global carbon cycles and thus in

climate change processes (Gruber et al., 2019). Furthermore, the SO is less affected by human

activities and so the influence of SSAs on CCN is thought to be particularly strong over this region.

     Cyclones can have great effect on marine aerosol, especially on SSAs in the atmosphere, as they

may transport large water volumes and at the same time impose strong winds (Fang et al., 2009).

Air convergence due to the reduction of pressure caused by cyclones may also affect SSA

concentration. There are typically many more cyclones developed during the summer time than

during other seasons in the study region. 959 cyclones occurred in the SO during the summer of

2004 to 2008 (Liu et al., 2012).

     In summary, the impact of cyclones on the emission of SSAs cannot be ignored, but it remains

unclear how changes in cyclone occurrence can impact the emissions of SSAs. Due to the threat

global warming poses, and the influence of CCNs, and hence SSAs, on related climatic processes,

it is particularly important and timely to consider this question although the lack of direct

observations makes this challenging. As cyclones develop in the westerlies and the SO, the decrease

of pressure, air convergence, strong winds and heavy precipitation may alter the emissions of SSAs



and CCNs and thus affect regional climate in the mid and high latitudes of the southern hemisphere.

The observation of cyclones is commonly performed at fixed points on land (Badarinath et al., 2008), but such observations cannot be used to investigate the effect of cyclones on SSAs over remote ocean regions. However, continuous observation technology now available on research vessels can greatly improve the temporal and spatial resolution of SSA particle measurements, allowing for the in situ characterization of marine SSA behavior in extreme weather.

In this study, SSA characteristics were observed with high temporal resolution during three cyclones in the SO to determine the transport of SSAs upward to the high atmosphere by cyclones. The concentration and particle size of SSAs were measured simultaneously for the first time at high-temporal resolution (1h) in the southern hemisphere mid and high latitudes during three cyclones (23 February to 4 March 2018). The results provide a new insight into the effect of cyclones on the

generation and vertical transport of SSAs in the mid and high latitudes of the SO.

## 2.    Methodology

### 2.1 Observational sites

Observations were carried out on board the R/V "Xuelong" during the 34th Chinese Antarctica Expedition Research Cruise. This survey covered with a large portion of the SO (40°S to 73°S,

170°E to 124°W, Fig. S1). The observations presented here were obtained between 23 February and 4 March 2018.

### 2.2 SSA measurement

Aerosol composition, including sea salts, were monitored at a temporal resolution of 1 h using an in situ gas and aerosol composition monitoring system (IGAC, Model S-611

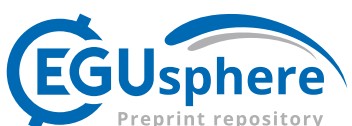

http://www.machine-shop.com.tw/). To minimize the impact of ship emissions, the sampling inlet

connected to the monitoring instruments was fixed 20 m above the sea surface on a mast located at

the bow of the research vessel. A total suspended particulate sample inlet was also positioned at the

top of the mast. All aerosol observational instruments were connected by conductive silicone tubing

with an inner diameter of 1.0 cm.

In the IGAC monitoring system, gases and aerosols were separated and streamed into liquid

effluent for on-line chemical analysis at an hourly interval (Young et al., 2016). In the sampling

process, fine particles were first enlarged by vapor condensation and subsequently collected on an

impaction plate. The samples were then analyzed for anions and cations using an on-line ion

chromatography system (DionexICS-3000). Six to eight concentrations of standard solutions were

used for calibration purposes, depending on the target concentration ($R^2$ values above 0.997). The

detection limits for $Na^+$ concentration was 0.03 μg $L^{-1}$ (aqueous solution).

**2.3 SSA particle size measurement**

A single particle mass spectrometer (SPAMS) was used to measure the SSA particle size

distribution. Details of the methods used for aerosol detection and the operational procedure for the

on-board SPAMS have been presented in a previous study (Li et al., 2014).

A PM2.5 collector was deployed to remove particles larger than 2.5 μm. Fine particles were

drawn into the vacuum system through a critical orifice and were then accelerated and focused to

form a particle beam. Particles with specific velocities then passed through two continuous diode

Nd:YAG lasers (532 nm). The aerodynamic diameter of a single particle was calculated using the

particle velocity. The particle sizes detected by the SPAMS were calibrated using polystyrene latex



spheres (PSL, Duke Scientific Corp., Palo Alto) with diameters of 0.2, 0.3, 0.5, 0.75, 1.0, 2.0, and

2.5 μm.

**2.4 Meteorological parameters**

Meteorological parameters such as wind speed (WS) and temperature were measured

continuously using an automated meteorological station mounted on the R/V "Xuelong". The time

resolution of the meteorological data is 1 hr. Weather map data, including sea surface pressure and

total precipitation, was obtained from the fifth generation ECMWF reanalysis for the global climate

and weather (ERA5. https://cds.climate.copernicus.eu/). Satellite cloud maps were obtained from

the Level-1 and Atmosphere Archive and Distribution System Distributed Active Archive Center

(LAADS DAAC data product MOD021KM. https://ladsweb.modaps.eosdis.nasa.gov/).

**2.5 Undisturbed SSA concentration estimates during the cyclone period**

Undisturbed SSA (U-SSA) concentrations during the cyclone period were estimated in the

following two ways.

The momentum flux at the air-sea interface, also called wind stress, is an important part of the

interaction between ocean and atmosphere and reflects the friction and drag effect between the two

fluids. Wind stress is the energy source of SSA formation. The momentum flux at the air-sea

interface can be calculated using the following equation (Toffoli et al., 2012):

$$\tau = \rho_a C_d U_{10}^2$$

where $\rho_a$ is the air density, $U_{10}$ is the wind speed as measured at 10 m above the mean water level,

and $C_d$ is a drag coefficient which can be expressed as follows:

$$C_d = (a + bU_{10}) \times 10^{-3}$$

where $a$ is 0.96 and $b$ is 0.06.



By means of comparing the difference of wind stress between cyclonic and non-cyclonic periods and combining with the concentration of SSAs during non-cyclonic periods to obtained U-

SSA $_{\text{(wind stress)}}$ concentration.

$$\text{U-SSA}_{\text{(wind stress)}} = \frac{\tau_{\text{cy}}}{\tau_{\text{non-cy}}} * \text{SSA}_{\text{(non-cy)}}$$

For the indirect production of SSAs through the formation and bursting of bubbles, the SSA flux function $dF_0/dr$ (particles m$^{-2}$ s$^{-1}$ mm$^{-1}$) which expresses the rate of sea water droplet generation per unit area of sea surface per increment of particle radius, is given by Monahan et al. (1986) as:

$$\text{SSA flux} = \frac{dF_0}{dr} = 1.373 U_{10}^{3.41} r^{-3} (1 + 0.057 r^{1.05}) \times 10^{1.19 e^{-B^2}}$$

where $B = (0.38 - \log r)/0.65$, $r$ is the particle radius and $U_{10}$ is as defined above.

U-SSA $_{\text{(Sea-salt flux)}}$ concentration can be obtained as follows:

$$\text{U-SSA}_{\text{(Sea-salt flux)}} = \frac{\text{SSA flux}_{\text{cy}}}{\text{SSA flux}_{\text{non-cy}}} * \text{SSA}_{\text{(non-cy)}}$$

**3. Results and discussion**

**3.1 Meteorology**

The observational campaign was carried out from 23 February to 4 March 2018. The study area is noted for the influence of cyclones and was defined by the outermost closed isobar surrounding the cyclone area center (Wernli and Schwierz, 2006). Rainfall affects numerous aspects of SSA production. Under certain conditions, raindrops falling onto the sea surface can produce SSA

particles directly as a result of their impact. However, falling raindrops can also function as efficient scavengers of particles in the atmosphere (Lewis and Schwartz, 2004). The effects of rainfall on SSA production are currently not well quantified. We therefore extracted data from periods of precipitation during the cruise. Three cyclone events were captured during this period (Fig. S2). Na$^+$ derived from SSA is an important component of marine atmospheric aerosols (Teinila et al., 2014)





and is generally considered to be a marker of SSAs in the marine atmosphere (Yeatman et al., 2001).

Hence, $Na^+$ concentrations and meteorological information from the study period are presented in

Fig. 1.

The R/V "Xuelong" started from 173°E 44°S and sailed due south, encountering a cyclone

which was generated at about 150°E, gradually moving eastwards (Fig. 2). As the cyclone

approached, the research vessel sampled a northwest warm and humid air mass followed by

precipitation. As the research vessel entered the cyclone area (Event 1. See shaded area, Fig. 1) at

about 15:00 24/2/18 (all times presented here are UTC), air pressure suddenly dropped (from 1003

hpa to 961 hpa) and wind speed significantly stronger than in the non-cyclone area was experienced

(average wind speed increased from 11.7 m $s^{-1}$ to 14.8 m $s^{-1}$).  However, the average $Na^+$

concentration during this cyclone event remained relatively constant as the WS increased, changing

from 1529 ng $m^{-3}$ to 1706 ng $m^{-3}$. At about 23:00 25/2/18, the research vessel left the cyclone area.

Note that wind speed dropped sharply between 13:00 and 23:00 25/2/18 (average 14.8 $ms^{-1}$ and 9.3

$ms^{-1}$, respectively) and this was matched by a rapid decrease in SSA concentrations (from 1706 ng

$m^{-3}$ to 343 ng $m^{-3}$. Fig. 1b).

As the vessel continued to move south it encountered another cyclone area at 10:00 26/2/18

and immediately turned to the southeast, leaving the cyclone area at 22:00 26/2/18 (Event 2. Shaded

area Fig. 1). During Event 2, the research vessel did not pass through the center of the cyclone.

However, the observed atmospheric pressure dropped from 983 hpa to 973 hpa and the average wind

speed increased from 13.5 m $s^{-1}$ to 15.5 m $s^{-1}$. Similar to Event 1, the average $Na^+$ concentration

during the cyclone period remained relatively constant, or even decreased (from 2810 ng $m^{-3}$ to

2354 ng $m^{-3}$), as the WS increased. The dominant air flow was cold and westerly thus there was



only a little precipitation (Fig. 2).

While the research vessel moved southeast and deep into the SO, Na$^+$ concentrations were much lower than during the first two cyclone events, which suggested that low air temperatures and sea ice cover reduces the production of SSAs (Fig. S3) (Yan et al., 2020). Between 18:00 1/3/18 and 04:00. 4/3/18, the research vessel encountered a third cyclone (Event 3). The wind speed increased from 7.5 m s$^{-1}$ to 21.5 m s$^{-1}$ and the air pressure dropped from 986 hpa at its highest to 960 hpa at its lowest. Similarly, the average Na$^+$ concentration during this cyclone period did not increase significantly (changed from 255 ng m$^{-3}$ to 335 ng m$^{-3}$). The cyclone encountered during Event 3 was relatively stable and moved slowly, and there was only a small amount of precipitation generated in the wind shear region.

**3.2 SSA concentration**

The variation of Na$^+$ concentrations in regions of different latitude are presented in Fig. S4 (Supplementary Material). Positive correlations between Na$^+$ concentrations and wind speeds were found in the low-middle latitudes (20°N- 40°S) (R=0.59, Fig.S4), where the change of atmospheric pressure was small (Fig. S5), implying that SSA generation was greatly influenced by the wind speed. However, correlations between Na$^+$ concentrations and wind speeds were relatively low in middle-high latitudes (40°S-60°S) and in the polar region (60°S-74°S) (R=0.45 and 0.05, respectively), where the change of pressure was highly variable (Fig. S5), suggesting that the cyclones may have affected the relationship between wind speed and SSA concentrations in the MBL, further resulting in climate effects.

To further investigate the influence of southern hemisphere middle and high latitude cyclones



on SSA concentrations, a non-cyclone "normal" period (5 and 6 April), which had stable pressure and relative humidity controlled by a constant air mass with no precipitation, was selected as a

control period. The relationship between WS and $Na^+$ concentrations under different meteorological conditions are illustrated in Fig. 3.

It is readily apparent that the $Na^+$ concentration, and hence the concentration of SSAs, increased as the WS increased during the control period (Fig. 3a, R = 0.74). Positive correlations between $Na^+$ concentrations and WS were also present during periods of non-cyclone during the three events (R

= 0.65, 0.64 and 0.50, respectively), which is in reasonable agreement with previous studies (e.g., O'Dowd and de Leeuw, 2007). It is worth noting here that the correlations between $Na^+$ concentrations and WS during Event 3 were lower than during the other two events. This may be due to the lower temperatures and sea ice cover there, weakening the influence of WS on SSA production (Yan et al., 2020).

In contrast, the correlations between $Na^+$ concentration and wind speed are much lower during all three cyclone periods (R = -0.32, 0.15 and 0.43) and precipitation periods (R = 0.08 and -0.02. Figs. 3b, 3c and 3d). During the cyclone periods, $Na^+$ concentrations changed irregularly as the WS increased, suggesting that rainfall was altering the influence of wind stress on SSA production. This implies that the effect of precipitation on the formation of SSA is complicated and that under this

circumstance, WS may not be the critical factor that affects SSAs emission. Further studies of how SSA concentrations change during precipitation periods are required.

During the cyclone periods which had strong WSs, we did not observe obvious correlations between WS and $Na^+$ concentration. Some $Na^+$ concentrations during cyclone periods were even lower than those during non-cyclone periods. This suggested that, although there were stronger WSs,





the low pressure caused by cyclones may transport a part of the SSA population upwards, producing

more CCN at higher altitudes and simultaneously reducing the concentrations of $Na^+$ in the MBL.

Another possibility is that the emission of SSAs during cyclonic periods may be lower than that in

non-cyclone periods inherently. This is discussed further below.

### 3.3 SSA particle size distribution

Generally, SSA generation increases with wind speed, however in this study it has been found

that higher wind speeds did not result in higher levels of SAA during cyclone conditions. It seems

that the generation of SAAs was suppressed during periods of cyclone. It is necessary to try to

determine whether the emission of SSAs in the cyclonic periods was higher than that in the non-

cyclone periods. Feng et al. (2017) reported that both SSA particle size and the concentration

increased with increasing wind speed. As the WS increased from 3.4 to 10 m s $^{-1}$, a 7–10 fold

increase in atmospheric salt concentration was observed. Log-normal distributions predict a 30-fold

increase in the concentration ($\mu g/m^3$) of particles larger than 1 x $10^{-9}$ g (10 μm radius) and a 50-fold

increase in the concentration of particles larger than 1 x $10^{-8}$ g (20 μm radius) (McDonald et al.,

1982). If the particle sizes of SSAs increase with increasing wind speed, this indirectly confirms

that the concentrations of SSAs also increase.

The size distributions of SSAs observed during the three cyclone events are presented in Table

S1 and Fig. 4. During Event 1, the difference between the number of SSA particles larger than 1.2

μm observed in cyclone and non-cyclone periods was about 11%. The change of SSA size

distribution during Event 2 and Event 3 were consistent with that during Event 1 (about 6% and 5%,

respectively). The size spectrum of SSA particles changed toward larger sizes during cyclone



periods during all three events. These result imply that cyclones in southern hemisphere middle and high latitudes enhance SSA generation. However, the increase of SSA concentrations was not present when high wind speeds occurred during cyclone periods, suggesting that SSAs may have been transported away from, or diluted in, the lower atmosphere.

**3.4 Estimation of the proportion of SSAs transported upward by cyclones**

The middle and high latitude SO and the Antarctic region is one of the most pristine in the world and serves as an important proxy for the pre-industrial atmosphere. Human activity has little impact here and anthropogenic aerosols account for a small proportion of the total aerosol population. Aerosols are typically derived from natural sources, including primary particles (sea

spray and bursting bubbles), which make up the vast majority of the aerosol mass. In this region cyclones tend to occur in summer, generating more SSAs due to high WS. However, our results suggest that air convergence caused by the cyclones may result in considerable quantities of SSAs being transported vertically to higher altitudes, which can partly explain why the mean number concentration of CCN/cloud droplets ($N_d$) in the SO in summer is much higher than that in winter

(Mycoy et al., 2020).

As mentioned above, the size of the sea salt particles were larger than no-cyclone period. However, the level of SSA hardly increased with the wind speed during the cyclone processes. It's likely that there is a part of SSAs been transported upward by air convergence due to cyclone. The transport effect of updraft on SSAs on the one hand reduces the concentration of SSAs on the sea

surface; on the other hand, a large number of SSAs are transported to the upper air, the SSAs reaching high altitudes can change the radiation reflected back to space by modulating the $N_d$, which





in turn changes cloud reflectivity even without any changes to cloud macrostructure (Twomey, 1977). Furthermore, by changing CCN/Nd, cloud microphysical processes are altered (Albrecht, 1989). These two effects, summarized in Fig. 5, can ultimately affect the radiation balance of the earth system in the mid and high latitudes of southern hemisphere (Quinn and Bates, 2011). Thus, the effect of cyclones on the production of SSAs, especially in the polar region, can not be neglected.

It is difficult to precisely estimate the proportion of SSAs transported vertically directly. However, the differences of wind stress or sea-salt flux between cyclone and non-cyclone periods can be calculated using the undisturbed concentrations of Na$^+$ (U-SSA concentration) during the cyclone period. This can be used to quantify the proportion of SSAs which may have been transported upward.

Fig. S6 and S7 shown the difference of wind stress and Sea-salt flux between cyclonic and non-cyclonic periods. The proportion of SSAs which are vertically transported, estimated using the wind stress method and the sea-salt flux method, are presented in Table 1. Using the wind stress method, more than 23.4% of the SSAs were transported upward by cyclone processes during Event 1, and 36.2% and 38.9% in Event 2 and Event 3, respectively. The proportion of SSAs transported upwards estimated using the bubble method were higher than those estimated using the wind stress method for all three events. The transported proportion during Event 3, when the research vessel was located closer to the Antarctic, estimated using the bubble method was the highest, reaching 56.6%, which was much higher than that estimated for Event 1 (39.9%) or Event 2 (42.8%).

The high transportation ratio of Event 3 agrees with the results of a previous study which reported that the largest contribution of SSA to CCN (up to 65%) was observed in the high southern latitudes (Quinn et al., 2017). There may have been another factor affecting Event 3 results, as the

research vessel was close to Antarctica where the WS was over 20 m s$^{-1}$ for a few hours. However,

as the sea state is typically not fully developed in such a situation, the energy flux from the air to

the ocean may differ from that under steady state conditions, as may wave breaking and hence SSA

production (Lewis and Schwartz, 2004). These circumstances can lead to the overestimation of the

proportion of SSAs which are vertically transported. In summary, the results suggest that over the

southern hemisphere middle and high latitudes, a significant proportion of SSAs are transported

upward and subsequently potentially effect climate change processes.

The influence of cyclone on SSAs in the tropics, where cyclones are much stronger and more

complicated, is not covered by this paper. Further studies of how SSA concentrations change in

tropical cyclone areas are required. However, the observational results presented in this study extend

the current knowledge of how cyclones influence marine aerosol emissions in the southern

hemisphere mid-and high-latitudes and their potential to alter climate change.

**Conclusions**

An underway aerosol monitoring system was used to determine the Na$^+$ concentration during

different cyclone periods in southern hemisphere mid and high latitudes in order to access the

potential effects of cyclone on SSA emissions. Three cyclones events were observed during the 34th

Chinese Antarctica Expedition Research Cruise from 23 February 2018 to 4 March 2018.

It was expected that the high wind speeds produced during the cyclone events should have

increased the generation of SSAs. However, although an increase in SSA particle size was observed,

there were no obvious SSA concentration increases during these periods. It is likely that there was

a proportion of the SSAs which were transported upward due to the cyclones. By means of assessing

the difference of wind stress and Sea-salt flux between cyclone and non-cyclone periods, It was

estimated that more than 23% of SSAs were transported upward during cyclone periods, with the highest proportion occurring in the SO (39% to 55%). Vertically transported SSAs can be regarded as an important source of CCN and hence have an effect on climate in the mid and high latitudes of the southern hemisphere.

The effects of cyclones on SSA emissions are indirect and complicated. Therefore, future work should be carried out on the effect of cyclones of varying intensity on SSA emissions and their generation mechanism during periods of precipitation, as well as their potential climate effects at the different latitudes.

**Acknowledgements**

This study is financially supported by the Qingdao National Laboratory for Marine Science and Technology (No. QNLM2016ORP0109), the Natural Science Foundation of Fujian Province, China (No. 2019J01120), the Response and Feedback of the Southern Ocean to Climate Change (RFSOCC2020-2025), the Chinese Projects for Investigations and Assessments of the Arctic and Antarctic (CHINARE2017-2020), and the National Natural Science Foundation of China (No. 41941014). The authors gratefully acknowledge the Guangzhou Hexin Analytical Instrument Company Limited for on-board observation technical assistance, and the Zhangjia Instrument Company Limited for IGAC technical assistance and data analysis.

**Data availability**

The data discussed in this manuscript are available from the following websites:
https://doi.org/10.5281/zenodo.7912911.

**Competing interests**

The authors declare that they have no conflict of interest.



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



**Figures/Tables**

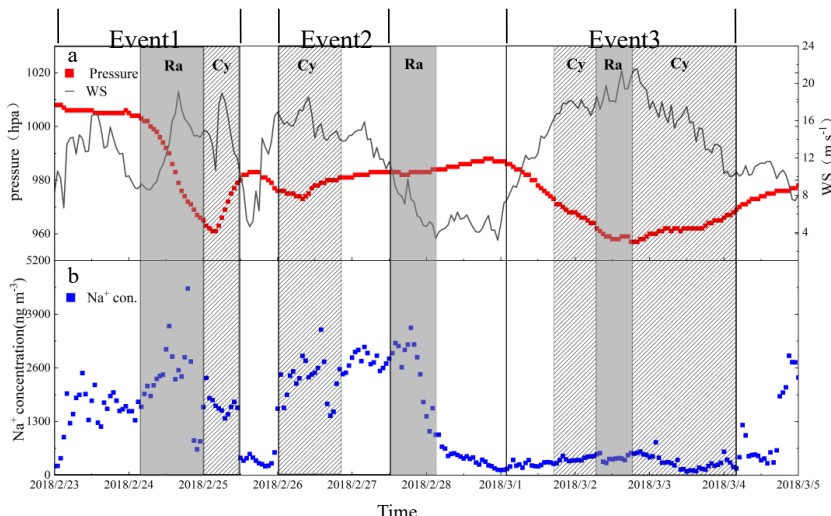

**Figure 1. Temporal distributions of Na$^+$ and relevant meteorological parameters obtained during the period 23 February to 4 March 2018 in the cyclone area of the Southern Ocean. (a) Time series of atmospheric**

**pressure (hpa) and wind speed (WS. m s$^{-1}$). (b) Time series of Na$^+$ concentrations (ng m$^{-3}$). Shading indicates: Ra - precipitation periods; Cy - cyclone periods. No shading corresponds to non-cyclone periods.**

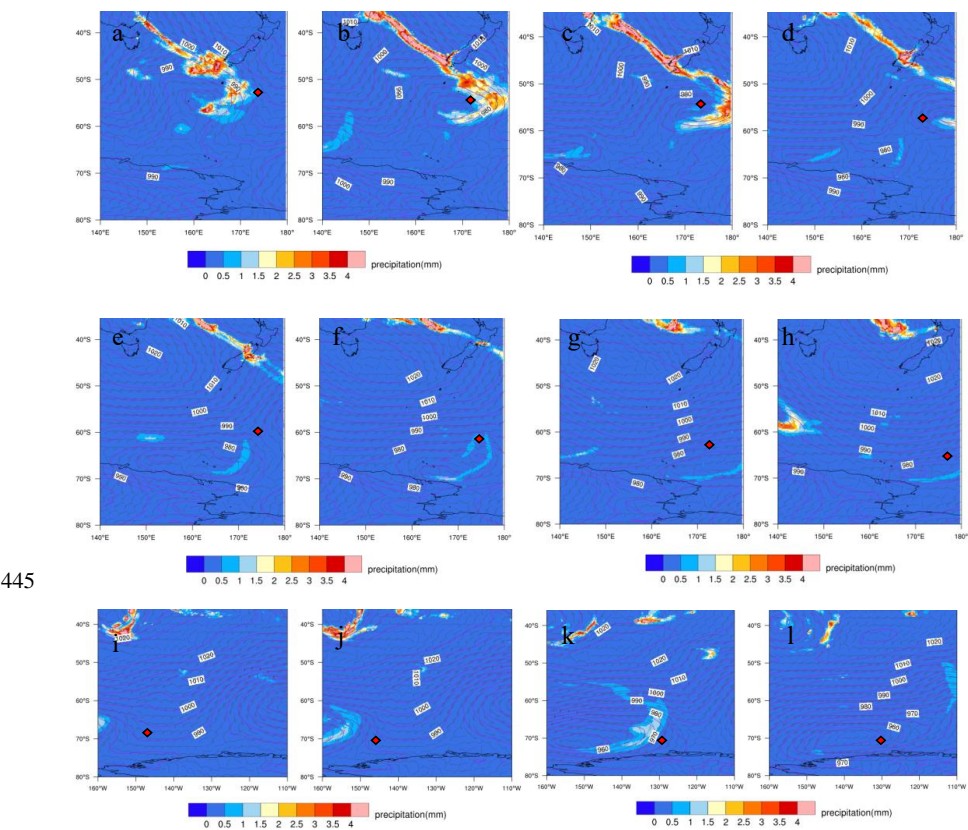

**Figure 2. Sea surface pressure (hpa) and total precipitation (mm) maps for the three observed cyclone events. Event 1: (a) 14:00  24/2/18; (b) 20:00 24/2/18; (c) 01:00 25/2/18; (d) 08:00 25/2/18. Event 2: (e) 16:00 25/2/18; (f) 00:00 26/2/18; (g) 20:00 26/2/18; (h) 18:00 26/2/18. Event 3: (i) 01:00 1/3/18 (j) 18:00 1/3/18 (k) 13:00 2/3/18; (l) 22:00 2/3/18. The red diamond represents the position of the research ship. All times are UTC. The coastline of Antarctica is seen at the bottom of each figure.**




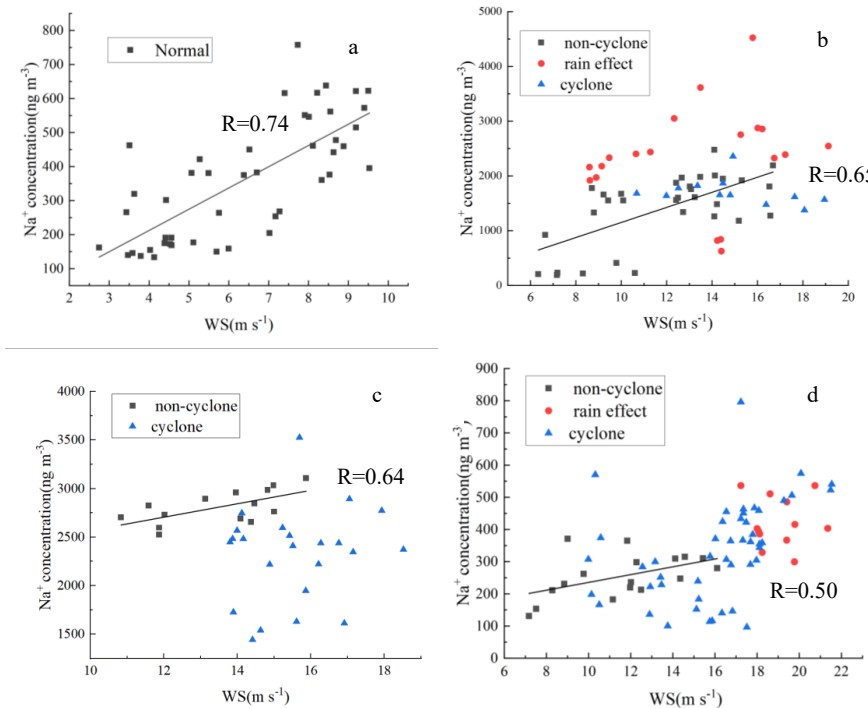

**Figure 3. Correlation between Na$^+$ concentration and wind speed under different meteorological conditions. (a) A non-cyclone "normal" period (i.e., stable air pressure and relative humidity, constant air mass, no precipitation). (b) Event 1 (cyclone, non-cyclone and raining periods). (c) Event 2 (cyclone and non-cyclone periods). (d) Event 3 (cyclone, non-cyclone and raining periods).**


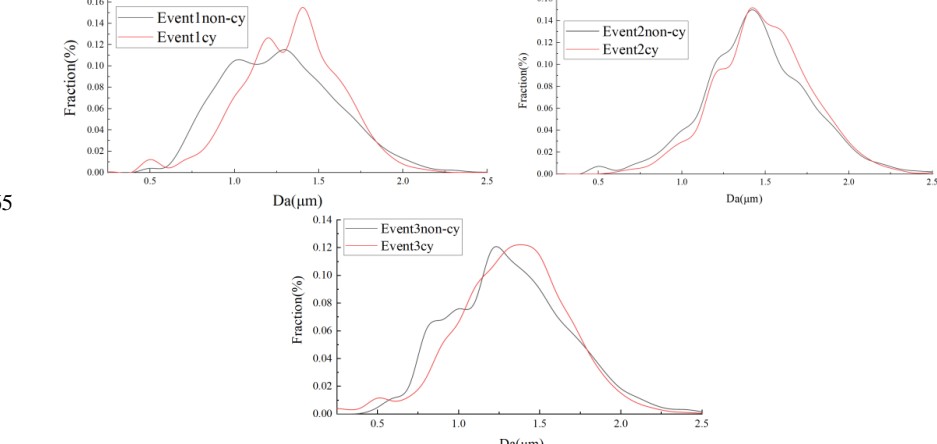

**Figure 4. SSA size distributions (in terms of fractional percent) for cyclone and non-cyclone periods during the three observed Events.**



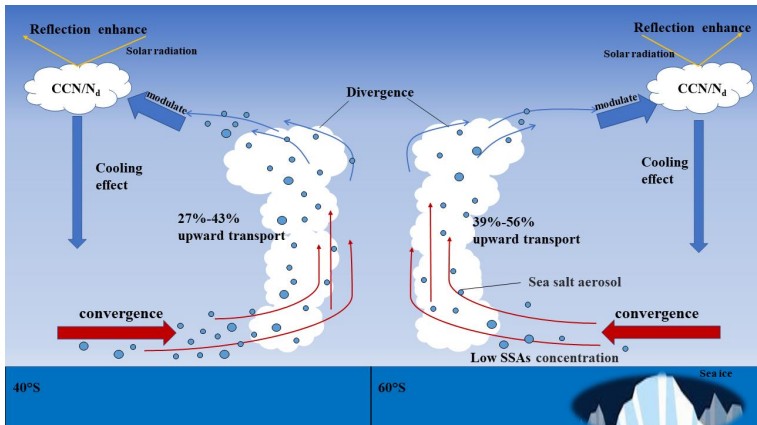


**Figure 5. Schematic diagram illustrating the impact of cyclone on SSA generation and transport and the resulting climate effects.**


**Table 1. Estimation of SSAs vertical transport proportion by assessing the difference of wind stress and Sea-salt flux.**

|  | Event 1 | Event 2 | Event 3 |
|---|---|---|---|
| Quotient of wind stress | 1.689 | 1.310 | 2.153 |
| Quotient of sea-salt flux | 2.156 | 1.463 | 3.031 |
| Average $Na^+$ con. (non-cy) $ng/m^3$ | 1273.19 | 2816.90 | 254.76 |
| Average $Na^+$ con. (cy) $ng/m^3$ | 1647.31 | 2353.74 | 334.94 |
| Estimated U-SSA(wind stress) con. $ng/m^3$ | 2151.05 | 3689.30 | 548.52 |
| Estimated U-SSA(Sea-salt flux) con. $ng/m^3$ | 2745.16 | 4113.74 | 772.14 |
| Estimated upward transport (wind stress) | 23.4% | 36.2% | 38.9% |
| Estimated upward transport (Sea-salt flux) | 39.9% | 42.8% | 56.6% |