# Peer review of "Cyclones enhance the transport of sea spray aerosols to the high atmosphere in the Southern Ocean"

_EGUsphere, 2023_

## Author Response (AR1)

**1) LIST OF RELEVANT CHANGES MADE IN THE MANUSCRIPT**

1. Abstract was improved following the reviewer's suggestions.

2. We modified the figures in the manuscript, increase the font sizes (Fig. 1, 3 and 4), add the correlation between wind speed and $Na^+$ during cyclone period (Fig. 3) and the moving direction of the research ship in Fig. S1.

3. SSA chemical composition measurement (IGAC and SPAM) was described in more detail (section 2.2 and 2.3).

4. Also in section 2.2, we describe in more detail about how to eliminate ship emission interference during sampling.

5.An additional part was included in Table S1, in order to show the details correlation about SSA chemical composition.

6. Wording in some sections (particularly in Abstract and Section 3) was modified, in order to reduce conjectures and improve reading. Typos in the manuscript have revised and the formulas in section 2.5 have been numbered. All these changes can be followed in the marked-up version of the manuscript.

7. Following the reviewer's suggestion, we used "sea spray aerosol" instead of "sea salt aerosol to make the relevant statements more reasonable.

8. Add the reference which statement "raindrops falling onto the sea surface can produce SSA"

9. All technical suggestions from both reviewers were included in the revised version.

**2)RESPONSE TO THE REVIEWERS' COMMENT**

**Response to Reviewer #1:**

Thank you for your comments. We have carefully reviewed the comments and revised the manuscript accordingly.

**[RC1]** The transport proportion was different during the 3 cyclones events, why?

Response: The maximum wind speed of event 1 and event 2 is less than 20m/s. Different to event 1 and event2, during the event 3, the research vessel was close to Antarctica where the WS was over 20 m s$^{-1}$ for a few hours. However, as the sea state is typically not fully developed in such a situation, the energy flux from the air to the ocean may differ from that under steady states, which may lead to the overestimation of thewind stress and SSA flux (formula 1 and 3), which ultimately caused the transport proportion in the event 3 is higher than other two events. (Lewis and Schwartz, 2004).

**[RC1]** SSAs mostly consists of NaCl and Mg, K, Ca sulfates. Details about SSA chemical composition measurement should be present, and explain the connection between them.

Response: Thank you for your suggestion. In fact, there are a significant correlation between $Na^+$, $Mg^{2+}$, $k^+$ and $Ca^{2+}$ in this study, specific results have been added to the revised manuscript.

**[RC1]** The influence of relative humidity on the change of particle size should be excluded.

Response: Relative humidity is higher on the ocean than on land which basically reached the

deliquescence point of NaCl(about RH 75%) (Cole et al, 2003). In this case, relative humidity has little influence on the change of particle size. We have revised in the manuscript.

**[RC1]** Why is the particle size change of Event 2 less significant than that of the other two events.

Response: Generally, SSA particle size increased with wind speed (Liu et al. 2020; McDonald et al., 1982). The change of wind speed in Event 2 is less significant than that of the other two events, which may explain the smaller change of particle size in event 2.

**[RC1]** As we know that the ship entertainments would influence the observation results, but how to exclude the ship emission in the study?

Response: Thanks very much for the comments. Shipboard observation is a challenge work in the marine environment. It is the case that ship emissions may impact the observation data. In this study, to minimize the impact of self-contaminations of the vessel on the observation results, the air inlet connecting to the monitoring instruments is fixed to a mast at 20 meters above the sea surface located at the bow of the R/V. Note that the major pollution source is from the chimney, which is located at the stern of the R/V and about 25 meters above the sea level. Hence, the pollution emissions from the vessel mainly located at the downwind of the sampling inlet, especially when the vessel is running forward. As high-time-resolution observations were used in this study, the self-contaminations from the vessel have been eliminated from the measurement results. The wind speed and wind directions were also monitoring during the observation period, which were used to determine if the observations were affected by the self-contaminations or not. The data have been corrected to eliminate the impact of ship contamination in this study.

**[RC1]** The description of aerosol measurement instruments should be added in the measurement section.

Response: Thank you for your suggestion, We have added the description of aerosol measurement instruments in the measurement section.

**[RC1]** The moving direction of the research ship needs to be added in Figure S1.

Response: We have added the moving direction of the ship in Figure S1.

**[RC1]** The value in Figure5 is inconsistent with the conclusion.

Response: There is an error in the Figure 5. We are very sorry about it. We have fixed the error in Figure 5.

**[RC1]** labels in Fig1 overlap with images.

Response: Thanks for the suggestion. We have corrected it in the Fig.1.

**[RC1]** It is suggested that U-SSA data in Table1 can be replaced with figure, which is more intuitive.

Response: Thanks for the useful suggestion. It is the case that figure is more intuitive than table. We have replaced the data in table1 with figure.

**[RC1]** There are some typos in the manuscript, please check and revised. (Such as "SAA")

Response: We are very sorry for the incorrect writing. The typos have been checked carefully and revised in the manuscript .

**[RC1]** Line55, rewrite this sentence to make it reasonable.

Response: We appreciate the reviewer's comment. We have rewritten the sentence in the manuscript.

**[RC1]** Line 127-128, "Undisturbed SSA (U-SSA)" need a more detailed definition.

Response: Undisturbed SSA (U-SSA) means the SSA concentration in the case of no cyclone disturbance which estimated by the differences of wind stress or sea-salt flux between cyclone and non-cyclone periods. We have added the detail explanation of 'Undisturbed SSA (U-SSA)' in the manuscript.

**[RC1]** Line 251, "The middle and high latitude SO and the Antarctic region is one of the most pristine in the world", Whether SO should be changed to Southern Hemisphere.

Response: Thanks very much for your suggestion. Southern Hemisphere has been accepted here.

**[RC1]** Line 234-239, More references are needed to support "particle sizes of SSAs increase with increasing wind speed".

Response: Thank you for your kind comment, we have added new references to support the statement that particle sizes of SSAs increase with wind speed.

**[RC1]** The formulas in section 2.5 should be numbered.

Response: We have revised in the manuscript. Thank you for your valuable suggestion.

**[RC1]** The correlation between wind speed and $Na^+$ during cyclone period also should be display on Figure 3.

Response: The correlation between wind speed and $Na^+$ during cyclone period have been calculated. The result has been included in the Fig. 3.

Reference:

Cole, I. S., Paterson, D. A., &Ganther, W. D. (2003). Holistic model for atmospheric corrosion Part 1 - theoretical framework for production, transportation and deposition of marine salts. Corrosion Engineering, Science and Technology, 38(2), 129–134.

Lewis, E. R., & Schwartz, S. E. (2004). Sea salt aerosol production: Mechanisms, methods, 648 measurements, and models. American Geophysical Union.

Liu, S., Liu, C. C., Froyd, K. D., Schill, G. P., Murphy, D. M., Bui, T. P., . . . Gao, R. S. (2021). Sea spray aerosol concentration modulated by sea surface temperature. Proc Natl Acad Sci U S A, 118(9).

McDonald, R. L., Unni, C. K., & Duce, R. A. (1982). Estimation of atmospheric sea salt dry deposition: Wind speed and particle size dependence. Journal of Geophysical Research, 87(C2), 1246.

Yong Yu, M. Liz Alexander, Veronique Perraud, Emily A. Bruns, Stanley N. Johnson, Michael J. Ezell, Barbara J. Finlayson-Pitts, 2009. Contamination from electrically conductive silicone tubing during aerosol chemical analysis, Atmospheric Environment,2836-2839.

Young, L.H., Li, C.H., Lin, M.Y., wang, B.F., Hsu, H.T., Chen, Y.C., Jung, C.R., Chen, K.C., Cheng, D.H., Wang, V.S., Chiang, H.C., Tsai, P.J., 2016. Field performance of semi-continuous monitor for ambient PM2.5 water-soluble inorganic ions and gases at a suburban site. Atmos. Environ. 144, 376–388.

**Response to Reviewer #2:**

Thanks very much for your comments. We have carefully reviewed the comments and have revised the manuscript accordingly.

**[RC2]** SPAMS cannot provide reliable size distribution measurements. It is well known that the transmission efficiency through SPAMS inlet and aerodynamic lens is highly size dependent, leading to a biased size distribution measurement.

Response: It is true that SPAMS cannot provide reliable size distribution measurements. With the SPAMS measurement, fine particles were drawn into the vacuum system through a critical orifice and were then accelerated and focused to form a particle beam. Particles with specific velocities then passed through two continuous diode Nd:YAG lasers (532 nm). The aerodynamic diameter of a single particle was calculated using the particle velocity (Li et al., 2011). In this case, the single particle size can be obtained by the fly distance and fly time. The particle size distribution can be obtained if sufficient sampling single particles were measured by SPAMS. A biased size distribution measurement of SPAMS determination may occur, but the particle size distribution measured by SPAMS can still represent the real particle size distribution. The

measurement of particle size distribution using SPAMS has been confirm in previous studies (Yan et al., 2016; Li et al., 2014; Ma et al; 2016). In this study, Thousands of single particles were measured during the different events. Hence, the size distribution measured by SPAMS in this study can reflect the variations of size distribution in cyclone and non cyclone periods.

[RC2] The efficient capture of SSA particles by raindrops should be quantitatively discussed.The production of sea spray aerosol due to raindrop impaction on the seawater surface also needs to be included in the discussion.

Response: Thanks very much for the comment. This is an interesting question. It is the case that raindrops falling onto the sea surface can produce SSA particles directly but can also function as efficient scavenge of particles in the atmosphere, which is hard to present the relationship between the raindrops and SSAs formation and scavenge of SSAs. Generally, the scavenge of fine particles by raindrops may be the major effect during the precipitation. In this study, we mainly focused on the effect of cyclone on SSAs formation and transport, hence, the data discussed in this study was during non-precipitation periods. The observation data during precipitation periods has been excluded. The effects of rainfalls on the formation and scavenge of SSAs will be further investigated in the near future.

[RC2] Details about SSA chemical composition measurement should be present. For example, what is IGAC? How does it measure gas and aerosol at the same time? Was a dryer placed before the SPAMS? It is extremely important to describe the experimental details properly.

Response: Thank you for making this valuable suggestion. The contents of SSA chemical composition determination have been further supplemented in the manuscript.

The IGAC monitoring system consisted of three main units, including a Wet Annular Denuder (WAD), a Scrub and Impact Aerosol Collector (SIAC), and an ion chromatograph with a sampling flow of 16.7 LPM. On the one hand, the collection of acidic and basic gases relies on the diffusion and absorption of gases into a downward flowing aqueous solution. and was positioned at an angle to facilitate the collection of enlarged particles. On the other hand, ultrapure water was fed continuously into the nozzle at 1.2 mL/min and heated to 140 °C to vaporize the water. Steam was sprayed directly towards the particle-laden air to improve the humidity of the flue gases. Fine particles were enlarged and subsequently accelerated through a conical-shaped impaction nozzle and collected on the impaction plate. The gas and aerosol liquid samples from the WAD and SIAC were drawn separately by a pair of syringe pumps, where one syringe collected the current sample (55 min) and the other injected the previous sample. The samples were then subsequently analyzed for anions and cations by an online ion chromatography (IC) system (Dionex ICS-3000). The injection loop size was 500 μL for both anions and cations.

A Nafion tube dryer was placed before the SPAMS to remove the moisture of sampling gas. We have added the description of experiment in detail in the manuscript.

[RC2] The figures are not well made. For example, some of the font sizes is small. The resolution seems to be low. And there should be a space between quantity and unit.

Response: Thank you for making this valuable suggestion. The figures have been revised in the manuscript.

Specific comments:

**[RC2]** I would use "sea spray aerosol" instead of "sea salt aerosol", as many recent studies identify that SSA contain a large fraction of organics.

Response: Thanks very much for your suggestion. "sea spray aerosol" was accepted instead of "sea salt aerosol" in the manuscript.

**[RC2]** Abstract: it is stated that "…has implications to climate change.". It is a very vague statement. I would suggest to be specific."

Response: Great comment, we have revised this section as "A large number of SSAs are transported to the upper air, the SSAs reaching high altitudes can change the radiation reflected back to space by modulating the $CCN/N_d$, which may affect the regional climate."

**[RC2]** Line 36: it is incorrect to state that SSA contains trances of organic materials. Instead, organics are the major component of SSA.

Response: Thank you for your comment. It's true that organics are very important component of sea salt aerosols. Here, the reference refer in particular to the "pure sea salt". We have revised "SSA"to "pure sea salt" in the manuscript.

**[RC2]** Line 70: why is global warming mentioned here?

Response: Sea spray aerosol can direct absorbing and scattering of solar radiation. Furthermore, Sea spray aerosol is an important source of CCN, which plays a significant role in regulating global warming, We have revised the relevant part of the manuscript.

**[RC2]** Line 95: how do the authors make sure that ship emission did not interfere the measurement?

Response: Thanks very much for the comments. The shipboard observation of atmosphere aerosol is great challenge in the marine environment. It is the case that ship emissions may impact the observation data. In this study, to minimize the impact of self-contaminations of the vessel on the observation results, the air inlet connecting to the monitoring instruments is fixed to a mast at 20 meters above the sea surface located at the bow of the R/V. Note that the major pollution source is from the chimney, which is located at the stern of the R/V. Hence, the pollution emissions from the vessel mainly located at the downwind of the sampling inlet, especially when the vessel is running forward. In addition, high-time-resolution observations are used in this study, the self-contaminations from the vessel have been eliminated from the measurement results. The wind speed and wind directions were also monitoring during the observation period, which were

used to determine if the observations were affected by the self-contaminations or not. The data have been corrected to eliminate the impact of ship contamination in this study.

**[RC2]** Line 97: silicone tubing should not be used for aerosol sampling for chemical analysis. As silicone could decompose and contaminate the aerosol samples.

Response: Generally, electrically conductive silicone tubing is used for aerosol monitoring system to minimize the loss of particles in the sampling tube (Yan et al., 2019). In some case, silicone could decompose and contaminate the aerosol samples, such as electrically conductive silicone tubing can be a significant source of siloxanes. Thus, silicone tubing may pose problems for analysis of trace species found in aerosols (Yong et al., 2009). In this study, To minimize the particle lost and residence time of the gases in the tubing, we used a high velocity sampling system, with a gas velocity of about 4.25 m/s in the tubing. The residence time of the gas sampling in the tubing was about 4.7 seconds, which can reduce the impact of silicone on aerosol samples. In this study, we focused mainly on the components of sea spray aerosols, which is without silicone. Hence, the electrically conductive silicone tubing used in this study has less effect on the observation results.

**[RC2]** Line 155: is there a reference to the statement "raindrops falling onto the sea surface can produce SSA"?

Response: Thank you for your question. We will refer to relevant articles in the manuscript. Previewer study has shown that the impact of hydrometeors (raindrops, hail, or SSA particles) on the ocean surface can produce SSA particles directly or indirectly, either from bubbles entrained by the drop or from bubbles or SSA particles produced by the splashed drops resulting from the original impact (Blanchard and Woodcock, 1957).

**[RC2]** Line 157: no, the scavenging of aerosol by rainfall has been well studied by many literatures.

Response: Thank you for your comments. The relevant statements in the article has been revised.

**[RC2]** Line 241: How was the size distribution of SSA measured? The size distribution from SPAMS is highly biased, due to the transmission efficiency of aerosols through its sizing region.

Response: A biased size distribution measurement of SPAMS determination may occur, but the particle size distribution measured by SPAMS can still represent the real particle size distribution. The measurement of particle size distribution using SPAMS has been confirm in previous studies (Yan et al., 2016; Li et al., 2014; Ma et al; 2016). With the SPAMS measurement, particles larger than 2.5 μm would be removed, fine particles were drawn into the vacuum system through a critical orifice and were then accelerated and focused to form a particle beam. Particles with specific velocities then passed through two continuous diode Nd:YAG lasers (532 nm). The aerodynamic diameter of a single particle was calculated using the particle velocity(Li et al., 2011). In this case, the single particle size can be obtained by the fly distance and fly time. The particle

size distribution can be obtained if sufficient sampling single particles were measured by SPAMS. In this study, Thousands of single particles were measured during the different events. Hence, the size distribution measured by SPAMS in this study can reflect the variations of size distribution in cyclone and non-cyclone periods.

**[RC2]** Figure 1: it is better to also include the rain rate data here.

Response: Thanks very much for the comment. Generally, the scavenge of fine particles by raindrops may be the major effect during the precipitation. In this study, we mainly focused on the influence of cyclones on SSAs formation and transport, hence, the data analyzed in this study was during non-precipitation periods. The effects of rainfalls on the formation and scavenge of SSAs will be further investigated in the near future.

Reference:

Blanchard, D.C.,and A.H. Woodcock, 1957. Bubble formation andmodification in the sea and its meteorological significance,Tellus,9,145-158.

Li, L., Huang, Z., Dong, J., Li, M., Gao, W., Nian, H., Fu, Z., Zhang, G., Bi, X., Cheng, P., Zhou, Z., 2011. Real time bipolar time-of-flight mass spectrometer for analyzingsingle aerosol particles. Int. J. Mass Spectrom. 303, 118e124.

Li, L., Li, M., Huang, Z., Gao, W., Nian, H., Fu, Z., … Zhou, Z. 2014. Ambient particle characterization by single particle aerosol mass spectrometry in an urban area of Beijing. Atmospheric Environment, 94, 323–331.

Ma, L., Li, M., Zhang, H., Li, L., Huang, Z., Gao, W., … Zhou, Z. 2016. Comparative analysis of chemical composition and sources of aerosol particles in urban Beijing during clear, hazy, and dusty days using single particle aerosol mass spectrometry. Journal of Cleaner Production, 112, 1319–1329.

Yan, J., Chen, L., Lin, Q., Zhao, S., & Zhang, M. 2016. Effect of typhoon on atmospheric aerosol particle pollutants accumulation over Xiamen, China. Chemosphere, 159, 244–255.

Yan, J., Jung, J., Zhang, M., Xu, S., Lin, Q., Zhao, S., Chen, L., 2019. Significant underestimation of gaseous Methanesulfonic Acid (MSA) over Southern Ocean. Environ Sci. & Tech. 53, 13064-13070.

Yong Yu, M. Liz Alexander, Veronique Perraud, Emily A. Bruns, Stanley N. Johnson, Michael J. Ezell, Barbara J. Finlayson-Pitts, 2009. Contamination from electrically conductive silicone tubing during aerosol chemical analysis,Atmospheric Environment,2836-2839.

Young, L.H., Li, C.H., Lin, M.Y., wang, B.F., Hsu, H.T., Chen, Y.C., Jung, C.R., Chen, K.C., Cheng, D.H., Wang, V.S., Chiang, H.C., Tsai, P.J., 2016. Field performance of semi-continuous

monitor for ambient PM2.5 water-soluble inorganic ions and gases at a suburban site. Atmos. Environ. 144, 376–388.

---

## Author Response (AR2)

*Editor decision: Publish subject to technical corrections*
*Comments to the Author:Dear Authors, I find that you have satisfactorily addressed reviewer concerns. My remaining concern is a series of English issues in the revisions that require more examination and correction before acceptance. Just a few examples (but not limited to these) include various issues in Lines: 112, 159, 232-234, 304, 308.*
*Please carefully go through the entire paper and conduct careful English editing.*

Thanks very much for your warm works and helpful comments. We have checked and revised the whole manuscript carefully, especially for the English writing and expressions. We have highlighted the changes in red in the revised manuscript.

*1. Line 112*

We have revised the sentence to "The collection of acidic and basic gases relies on the diffusion and absorption of gases into a downward flowing aqueous solution. The SIAC was positioned at an angle to facilitate the collection of enlarged particles.".

*2. Line 159*

We have changed the font in the formula to match the main text in revised manuscript.

*3. Line 232-234*

We have revised the whole sentence to "To further investigate the influence of the cyclone on SSA concentrations in the mid- and high Southern Hemisphere, a non-cyclone period (April 5th and 6th) with stable pressure and relative humidity but without precipitation, was selected as a control period (defined as normal period)." In the revised manuscript.

*4. Line 304*

We have revised in the manuscript.

*5. Line 308*

We have revised in the manuscript.

6. The corrections in the revision manuscript were not listed here, but they were highlighted in red.

7. We have revised the reference list to consist with the standards format.

8. We have added the part "Author contributions" in the manuscript.

9. We have modified the issue of font mismatch between the text and other formulas.

**The following is modified version of the manuscript**

[revised manuscript text omitted]